# Optimization of *Mangifera indica* L. Kernel Extract-Loaded Nanoemulsions via Response Surface Methodology, Characterization, Stability, and Skin Permeation for Anti-Acne Cosmeceutical Application

**DOI:** 10.3390/pharmaceutics12050454

**Published:** 2020-05-14

**Authors:** Worrapan Poomanee, Watcharee Khunkitti, Wantida Chaiyana, Pimporn Leelapornpisid

**Affiliations:** 1Department of Pharmaceutical Sciences, Faculty of Pharmacy, Chiang Mai University, Chiang Mai 50200, Thailand; worrapan.p@cmu.ac.th (W.P.); wantida.chaiyana@cmu.ac.th (W.C.); 2The Biofilm Research Group, Faculty of Pharmaceutical Sciences, Khon Kaen University, Khon Kaen 40002, Thailand; watkhu@kku.ac.th; 3Innovation Center for Holistic Health, Nutraceuticals, and Cosmeceuticals, Faculty of Pharmacy, Chiang Mai University, Chiang Mai 50200, Thailand

**Keywords:** nanoemulsions, response surface methodology, *Mangifera indica* L., statistics, permeability, *Propionibacterium acnes*

## Abstract

This study aimed to optimize nanoemulsions loading with *Mangifera indica* L. kernel extract using response surface methodology for enhancing the stability and skin permeation of the extract. Central composite design was employed for optimization and evaluation of three influencing factors including hydrophile-lipophile balance (HLB), % co-surfactant (PEG-7 glyceryl cocoate), and surfactant-to-oil ratio (SOR) on physical properties of the nanoemulsions. The desired nanoemulsions were then incorporated with the extract and characterized. Physicochemical properties of the extract-loaded nanoemulsions and their antibacterial effects against *Propionibacterium acnes* were also evaluated after storage at various conditions and compared to those of the initial. Ex vivo skin permeation was also investigated. The factors significantly (*p* < 0.05) influenced on droplet size, polydispersity index (PDI), and zeta potential, especially HLB of the surfactant and its combined effects with co-surfactant and SOR. The extract-loaded nanoemulsions revealed a very small spherical droplets (size of 26.14 ± 0.22 nm) with narrow size distribution (PDI of 0.16 ± 0.02). The formulation also presented an excellent stability profile and successfully enhanced antibacterial stability of the extract comparing with the extract solution. Ex vivo skin permeation study illustrated that the extract in nanoemulsions could be delivered through a primary skin barrier to reach viable epidermis dermis layers. In conclusion, the affinity of surfactant and hydrophilicity of the system play a crucial role in nanoemulsions’ characteristics. Such results might provide promising anti-acne nanoemulsions with the notable capacities of extract stabilization and permeation enhancing which will be further clinically evaluated.

## 1. Introduction

Acne is one of the greatest dermatological concerns which currently affects approximately 9.4% of the global population [1]. Not only physical symptoms as soreness and itchy in which acne results, but also psychological issues such as self-esteem and confidence are degraded. Acne inflammation is mainly attributed to the immunomodulatory property of *Propionibacterium acnes*, the Gram-positive anaerobe mainly residing within human pilosebaceous units [2,3]. Typically, topical antibiotics such as clindamycin and erythromycin are prescribed to treat mild to moderate acne in concomitant with other topical treatments [4]. Unfortunately, the antibiotic resistance of *P. acnes* highly occurs worldwide [4,5,6]. As a result, alternative anti-acne products especially those from natural resources have been well utilized in terms of co-presciption as well as over-the-counter (OTC) products which were reported to be safe and effective [7,8].

Nanoemulsions, one of the lipid-based nanocarriers, considerably gains the attention in both pharmaceutical and cosmetology points of view [9]. As a typical droplet size range of 20–200 nm, nanoemulsions present several advantages above other conventional formulations including optically transparent and the good kinetic stability due to an absence of gravitational separation [10]. Importantly, this nano-sized colloidal dispersion presents a variety of potential properties for enhancing skin permeation and stability of the loaded components [11,12]. Currently, nanoemulsions can be fabricated through using two methods as high energy and low energy emulsifications. High energy emulsification is a conventional method using a high-pressure homogenizer, sonicator, or microfluidizer [13,14]. On the contrary, low energy emulsification relies on the fact that nanoemulsions are possibly produced using low input energy under certain system conditions and compositions [14]. Particularly, phase inversion temperature (PIT) method was substantially employed to fabricate the nanoemulsions due to the capacity of large-scale manufacture without any requirement of complicated instruments [15,16,17]. Although the nanoemulsions produced by PIT method have been widely studied, nonetheless, no previous study has demonstrated the optimizing approach for the low energy procedure through the use of response surface methodology (RSM) which serves as a statistical method providing a reduction in the number of experiments for evaluating many variables. The development of these statistic models potentially indicated the significance of the independent variables and the investigation of synergistic or antagonistic interactions between different studied variables [18,19].

*Mangifera indica* L., categorized in genus *Mangifera*, which belongs to a plant family of Anacardiaceae, has been widely grown in tropical regions, especially India and Thailand [20,21,22]. The kernel of *M. indica* has been utilized as an astringent, vulnerary agent historically due to its perceived medicinal vitalities such as antibacterial, anti-inflammatory, analgesic, free-radical scavenging, and antioxidant activities since it contained high polyphenol content [23,24,25,26]. Our previous study also denoted that the ethanolic fraction of *M. indica* kernel (Kaew-Moragot cultivar), particularly grown in the North of Thailand, is a potential anti-acne agent due to its notable antibacterial activity against *Propionibacterium acnes* as well as its potent free-radical scavenging and inhibitory effects on acne-related pro-inflammatory cytokines [3]. However, the application of the natural compound, especially *M. indica* kernel extract in a sort of solution is limited due to its instability during storage, poor aqueous solubility as well as less skin permeability [27]. Therefore, our study aimed to optimize the oil-in-water nanoemulsions using RSM and to develop nanoemulsions loading the ethanolic fraction of *M. indica* kernel (Kaew-Moragot cultivar) in an attempt to enhance the stability and skin permeability of the extract to be a promising anti-acne product.

## 2. Materials and Methods

### 2.1. Plant Materials

*M. indica* kernels (Kaew-Moragot cultivar), derived from raw fruit, were supplied from a mango processing factory in Lumphun Province, Thailand during the period of May–September. Firstly, the dried kernels were mashed into fine powder and stored at room temperature until further extraction.

### 2.2. Preparation of M. indica Kernel Extract

The procedure of our previous study was followed [3]. Briefly, the powdered kernel was de-waxed using hexane. The plant residue was then fractionally macerated using ethyl acetate for 48 h, three cycles. The plant residue from ethyl acetate fractionation was then macerated by 95% ethanol for 48 h, three cycles. Ethanol was totally excluded through evaporation using rotary evaporator (Buchi^®^ Rotavapor R-300, Bangkok, Thailand) to produce an ethanolic fraction (EF) as a dark brown semi-solid extract with a yield of 12.12 ± 1.19% *w/w* dry weight. The EF was stored in tightly closed vials at 4 °C until use.

### 2.3. Preparation of Nanoemulsions

The mixture of an oil phase, composed of safflower oil, PEG-7 glyceryl cocoate, the surfactant mixture of Ceteareth-20, PEG-40 hydrogenated castor oil and Sorbitan oleate along with Butylated hydroxytoluene (BHT) was mixed until homogenous and heated to 80 ± 2 °C, which was higher than the PIT of the surfactant mixture that was previously determined by conductivity measurement, as 78.5 °C. An aqueous phase, which consisted of PEG-400 and DI water were also heated to 85 ± 2 °C and then added into the oil phase with continuous stirring using high speed homogenizer (IKA^®^ T25 Digital Ultra Turrax^®^, Staufen, Germany) at 7000 rpm to form a primary emulsion. Subsequently, the emulsion was rapidly cooled down using ice bath with constant stirring to form nanoemulsions.

### 2.4. Experimental Design

In order to evaluate the effects of hydrophile-lipophile balance (HLB) value of the surfactant mixture (*X*_1_), the percent of PEG-7 glyceryl cocoate (*X*_2_) and surfactant-to-oil ratio (SOR; *X*_3_) on three response variables including droplet size (*Y*_1_: nm), Polydispersity index (PDI; *Y*_2_), and zeta potential (*Y*_3_: mV), a three-factor central composite design (CCD) was performed due to its high accuracy for estimating factor effects and ability for determining interaction effects between factors [9]. The experimental results were statistically analyzed using analysis of variance (ANOVA) and RSM to optimize the statistical significances of model terms and the interaction of the independent variables. The code levels of independent variables, shown in a quadratic model and the matrix of CCD for RSM, are presented in Table 1 and Table 2, respectively. Table 3 shows the proportion of all surfactants in the surfactant mixture which was used in variation of HLB value.

The desirable characteristics of the response variables as minimum droplet size (*Y*_1_), minimum PDI (*Y*_2_), and optimum zeta potential (*Y*_3_), were predicted using RSM to forecast an adequate condition of the independent variables. The response surface models for describing the variation in response variables of droplet size (*Y*_1_) and PDI (*Y*_2_) are shown as following second-order polynomial Equation (1) whereas that of zeta potential (*Y*_3_) is shown as a following inverse Equation (2):(1)$$Y_{i}=\beta_{0}+\sum \beta_{i}\;X_{i}+\sum \beta_{ii}\;{X^{2}}_{i}+\sum \beta_{ij}\;X_{i}\;X_{j}$$
(2)$$\frac{1}{Y_{i}}=\beta_{0}+\sum \beta_{i}\;X_{i}+\sum \beta_{ii}\;{X^{2}}_{i}+\sum \beta_{ij}\;X_{i}\;X_{j}$$
where *Y_i_* is the predicted response value; β*_0_* is a constant; β*_i_*, β*_ii_*, and β*_ij_* are linear, quadratic, and interaction regression coefficients, respectively. An adequate mathematical model, which shown the significant term (*p* < 0.05), non-significant lack of fit, and high coefficient of determination (*R*^2^), was provided by Design-Expert^®^ software (version 7.1; Stat-Ease Inc., Minneapolis, MN, USA). The *R*^2^ value of a good fitness model should be at least 0.80 [9]. The non-significant variable (*p* > 0.05) was excluded from the final reduced model. Unless a quadratic or interaction term of the variable was significant (*p* < 0.05), the linear term was then kept in the final reduced model.

### 2.5. Optimization and Verification Procedures

In order to visualize the interaction effect of the independent variables on the responses, a three-dimensional response surface plot was created according to the final reduced model. Two variables, shown an interaction effect on the response variable, were varied within the experimental range while one variable was set at the constant point. The numerical optimization was also performed to optimize adequate values of the independent variables with desirability of 1. Additionally, the final reduced models were verified by quantitative comparison between the actual experimental and theoretical predicted results. Percentage of prediction error between these results should be not more than 5% to accept the adequate statistical models.

### 2.6. Formulation of the Extract-Loaded Nanoemulsions

The selected nanoemulsions were then loaded with the ethanol fraction extract of *M. indica* kernel (EF) dissolved in PEG-400 in the concentration of 2.5% *w*/*w*. The surfactant system consisted of ceteareth-20, PEG-40 castor oil and sorbitan oleate in a ratio of 1:2:2 (HLB = 10). The EF-loaded nanoemulsions with the surfactant concentrations of 9.0% and 9.5% were named as N1-EF and N2-EF, respectively. The N1-EF and N2-EF were prepared by the PIT method as previously described in Section 2.3.

### 2.7. Determination of Droplet Size, Polydispersity Index (PDI), and Zeta Potential

Nanoemulsions were 100-fold diluted by DI water before the measurement to avoid multiple scattering. The measurements of droplet size, PDI, and zeta potential were carried out using a photon correlation spectroscopy (Zetasizer^®^, Malvern, UK) at 25 ± 1 °C. The experiments were done in triplicate and the results were expressed as mean ± SD. Droplet size, PDI, and zeta potential of the EF-loaded nanoemulsions (N1-EF and N2-EF) were compared with those of their unloaded nanoemulsions (N1 and N2).

### 2.8. Transmission Electron Microscopic Analysis

The morphology of the internal droplets of the EF-loaded nanoemulsions was visualized by TEM. The nanoemulsions were 100-time diluted by DI water and applied onto 300-mesh copper grid. Each grid was added with a drop of 1% phosphotungstic acid to dye droplets and left at room temperature for 16 h. The ultrastructural image was visualized using TEM; EOL JEM-1200 EXII (Japan) at 80 kV at 40,000× magnification.

### 2.9. Stability Study

#### 2.9.1. Storage Conditions

The EF-loaded nanoemulsions (N1-EF and N2-EF) and the EF-solution (2.5% *w/w* EF dissolved in PEG400) were stored in well-tight containers under various conditions including 4 °C, room temperature with light (RTL), room temperature without light (RTD) and 45 °C/75% humidity for 90 d according to COLIPA guideline (2004), along with six cycles of heating-cooling (HC 1 cycle; 4 °C for 48 h, followed by 45 °C for 48 h).

#### 2.9.2. Physical Stability

The sedimentation, color, and phase separation were observed. Physical characteristics as droplet size, PDI, and zeta potential of N1-EF and N2-EF were evaluated after storage and compared with those of the initial point.

#### 2.9.3. Chemical Stability

Chemical stabilities of N1-EF, N2-EF, and EF-solution after storage were investigated by determination of gallic acid content which is its main compound using high performance liquid chromatography (HPLC). N1-EF and N2-EF were dissolved in absolute ethanol in a ratio of 1:1 and mixed by vortex mixture for 1 min by which EF was extracted from the nanoemulsions. The mixtures were then sonicated for 30 min by a sonicator (Elma^®^ S 30 H, Elmasonic, DKSH, Bangkok, Thailand) and centrifuged at 10,000 rpm by a centrifuge (Thermo Scientific^®^ Sorvall ST 16R, Bangkok, Thailand), at 15 °C for 30 min. The supernatant was collected for analysis. The method of HPLC determination was mentioned in the study of Poomanee et al. (2018). The amount of gallic acid in the formulation (mg/L) was calculated using the following Equation (3), which was provided by gallic acid calibration curve:*G* = 68.54*X*−12.08; *R*^2^ = 0.9994(3)
where *G* is AUC (mAU) and *X* is gallic acid concentration (mg/L). Percentage reduction (%) of gallic acid from the initial after storage was then calculated using the following Equation (4);
(4)$$\mathrm{Percentage}\;\mathrm{reduction}\;(\%)=\frac{GS-GD\;0}{GD\;0}\times 100$$
where *GS* is the amount of gallic acid after storage (mg/L) and *GD* 0 is the amount of gallic acid at D 0 (mg/L).

#### 2.9.4. Stability of Antibacterial Activity against *P. acnes*

Antibacterial activities of N1-EF, N2-EF and EF-solution against *P. acnes* were investigated using a broth microdilution method as mentioned in the study of Poomanee et al. [3]. Minimum inhibitory concentration (MIC) is regarded as the lowest concentration that shows no visible growth of the tested organisms. The *P. acnes* suspension, cultured in broth served as a bacterial control, while the mixture of broth and extract without *P. acnes* served as a negative control. Solvent control as 25% ethanol in broth was also evaluated. Subsequently, an aliquot (5 µL) of the bacterial suspension and sample (at MIC) was inoculated onto thioglycollate agar and incubated at 37 ± 1 °C, under anaerobic condition for 72 h. The lowest concentration, which completely defeats the bacterial population, is recognized as minimum bactericidal concentration (MBC). MIC and MBC of each sample after storage were compared with those of the initial point of each group.

### 2.10. Ex Vivo Skin Permeation Experiment

The stable formulation was selected for ex vivo skin permeation study. The procedure was performed following the method of Chaiyana et al. [28] with slight modifications. Skin was obtained from the flank area skin of three different stillborn piglets (Sus scrofa, Duroc), which were freshly obtained from faculty of veterinary medicine, Chiang Mai University, Thailand. Franz diffusion cells were then employed with an effective diffusional area of 2.46 cm^2^ and compared to those of the extract solution. Each sample (1 g) was applied onto the skin surface in the donor compartment. Aliquots of the receptor medium (1 mL) were collected at 1, 2, 4, 6, and 8 h for determining the amount of gallic acid using HPLC analysis as mentioned earlier. Percentage accumulation of gallic acid (*A*%) at each sampling interval was subsequently calculated using the following Equation (5):(5)$$A\%=\frac{\mathrm{Amount}\;\mathrm{of}\;\mathrm{gallic}\;\mathrm{acid}\;\mathrm{in}\;\mathrm{receiver}\;\mathrm{compartment}}{\mathrm{Amount}\;\mathrm{of}\;\mathrm{gallic}\;\mathrm{acid}\;\mathrm{in}\;\mathrm{sample}\;1\;g}\times 100$$

The amounts of gallic acid retained in different skin layers as stratum corneum (SC) and viable epidermis dermis (VED) as well as RC after 8 h were quantified using the following Equation (6) and expressed as cumulative amounts of gallic acid through the skin per unit surface area (*Q*_8_: µg/cm^2^).
(6)$$Q_{8}=\frac{\mathrm{concentration}\;\left(µg/\mathrm{mL}\right)\times \;\mathrm{dilution}\;\mathrm{factor}}{\mathrm{surface}\;\mathrm{area}\;\mathrm{of}\;\mathrm{skin}\;\left({\mathrm{cm}}^{2}\right)}$$

### 2.11. Statistical Analysis

All experiments were performed in triplicate. The results were expressed as mean ± SD. SPSS version 17.0 software (IBM co. Ltd., Armonk, NY, USA) for Windows was performed for statistical analysis characterization, stability, and ex vivo skin permeation. The differences of characterization, stability, and ex vivo skin permeation among the formulations were evaluated by one-way ANOVA with multiple comparisons using Tukey test, while the differences from D 0 of each formulation in stability testing were evaluated using paired *t*-tests. *p* values less than 0.05 (*p* < 0.05) were recognized as statistically significant.

## 3. Results

### 3.1. Optimization of Nanoemulsions and Data Analysis

The experimental results of the three response variables as shown in Table 2 were statistically analyzed to establish the best fit model. The obtained models exhibited high coefficient of determination (*R*^2^) in the range of 0.836–0.938 which significantly fitted for all response variables (Table 4). Therefore, *R*^2^ of the obtained models (Table 4) indicated that more than 83% of the response variations could be explained by RSM models. In addition, as shown in Table 5, a high *F*-value and a small *p*-value of each term means a highly significant effect on the response variables.

#### 3.1.1. Droplet Size

As shown in Table 4, the droplet size decreased by increasing of HLB and PEG-7 glyceryl cocoate, which was consistent with the results shown in Figure 1a. Besides, the droplet size decreased due to the increasing of SOR as shown in Figure 1b and Table 4.

#### 3.1.2. Polydispersity Index (PDI)

Table 4 and Table 5 revealed that PDI (*Y*_2_) was negatively proportional to the significant (*p* < 0.05) main effects of HLB along with the interaction effect of HLB and PEG-7 glyceryl cocoate. The response surface plot for the significant (*p* < 0.05) interaction effect of HLB and PEG-7 glyceryl cocoate on PDI also confirmed that PDI decreased due to the increasing of HLB and PEG-7 glyceryl cocoate as shown in Figure 1c which was in correspondence with the results of droplet size. PDI was also negatively proportional to the significant (*p* < 0.05) main effects of SOR as shown in Table 4 and Table 5, which indicated that with the increasing of surfactant concentration, PDI was subsequently decreased.

#### 3.1.3. Zeta Potential

As shown in Table 4 and Table 5, the increasing of HLB and surfactant concentration contributed to the increasing of zeta potential, which was consistent with the results of Figure 1d.

From the results, the safflower oil-based nanoemulsions showing very small droplet size and narrow size distribution could be fabricated using PIT method with the specific compositions and environments as high HLB value (10.0), high surfactant concentration along with an adequate amount PEG-7 glyceryl cocoate (2%). Therefore, EF-loaded nanoemulsions were prepared using these conditions and evaluated for the stability and ex vivo skin permeability.

### 3.2. Optimization of Responses for Formulating Safflower Oil-Based Nanoemulsions

For optimum safflower oil-based nanoemulsions with small droplet sizes of 20 nm, the lowest PDI with an optimum zeta potential value can be fabricated using the HLB 10.0 surfactant system, 2% *w*/*w* PEG-7 glyceryl cocoate, and the SOR of 1.9:1 representing the surfactant concentration as 9.5% *w*/*w*. Based on the optimum nanoemulsions, the predicted values of droplet size, PDI, and zeta potential obtained from the numeric optimization are 20 nm, 0.12, and −20.47 mV, respectively.

The actual experimental and theoretical predicted results were statistically compared as shown in Table 6. Percentage prediction errors of all responses were less than 5% which indicated whether the final reduced models were acceptable.

### 3.3. Formulation and Characterization of Nanoemulsions and EF-Loaded Nanoemulsions

The N1-EF and N2-EF were pale yellow in color with transparent optical characteristics due to their very small droplet sizes of 26.82 ± 0.28 nm and 26.14 ± 0.22 nm, respectively. The characteristics of N1-EF and N2-EF in comparison with their plain nanoemulsions (N1 and N2) are shown in Figure 2. The results also indicated that a lower surfactant concentration of N1 (9% surfactant) gave a significantly higher PDI value than N2 (9.5% surfactant). The zeta potential values of N1-EF and N2-EF were not significantly different from those of N1 and N2. The ultrastructural image of EF-loaded nanoemulsions (N2-EF) is shown in Figure 2d. The droplets were small spherical shapes.

### 3.4. Stability Study

After 90 d of storage, all nanoemulsions presented a good physical stability without phase separation and the increasing of turbidity except N1-EF. The droplet size, PDI, and physical appearances of N1-EF and N2-EF after storage are shown in Figure 3 comparing with those of D 0. Droplet size and PDI of N1-EF significantly increased after storage at 45 °C and six cycles of HC together with an increasing of turbidity (Figure 3). Figure 4 illustrated that zeta potential of N1-EF after storage in all conditions and that of N2-EF after storage at 45 °C presented significant differences from D 0.

The extract loaded-nanoemulsions were determined for their chemical and antibacterial stabilities which were also compared to the extract-solution (EF-solution) in order to evaluate the ability of stability enhancement of nanoemulsions. Chemical stabilities of the formulations are shown in Figure 5. EF-solution showed the highest percentage reduction in all storage conditions while the highest percentage reductions of N1-EF and N2-EF only occurred after storage in six cycles of HC.

Additionally, the EF-encapsulated nanoemulsions exerted the antibacterial effect against *P. acnes*. After storage, no changes of MIC and MBC against *P. acnes* of all formulations were observed (Table 7). On the contrary, MIC and MBC of EF-solution apparently increased after storage, which was in correspondence with the chemical stability of gallic acid within EF-solution as aforementioned. It was noteworthy that MIC of EF-solution at the initial was 2-fold lower than that of the extract-loaded nanoemulsions.

### 3.5. Ex Vivo Skin Permeation

As shown in Figure 6a, percentage accumulative of gallic acid in receiving chamber (RC) of N2-EF was the highest comparing with EF-solution. Cumulative amount of gallic acid per unit area of skin after 8 h of application (*Q*_8_) as shown in Figure 6b presented that *Q*_8_ of N2-EF in RC and VED was higher than that of EF-solution indicating permeation enhancing property of the nanoemulsions.

## 4. Discussion

Optimizing the nanoemulsions formulation, the experimental design was carried out based on previous information defining the important factors affecting to PIT method. The PIT method relies on changes of temperature which alters an affinity of thermal-sensitive non-ionic surfactants particularly polyethoxylated surfactants, for hydrophilic and lipophilic phases. In this study, the mixture of non-ionic surfactants composed of ceteareth-20, PEG-40 hydrogenated castor oil, and sorbitan oleate was employed. At low temperature, the surfactants are relatively hydrophilic due to positive curvature of the surfactant monolayer which contribute to the formation of oil-in-water emulsions. By increasing temperature, the oxyethylene groups of the surfactants are dehydrated leading to be relatively lipophilic, because of negative curvature promoting the formation of water-in-oil emulsions [10,15]. At PIT point, the affinity of the surfactant is equal for hydrophilic and lipophilic phases leading to the formation of bicontinuous microemulsion where the ultra-low surface tension appears. Thus, oil-in-water nanoemulsions which show very small droplets could be fabricated by low input energy at this point with enough surfactant molecules and the subsequent rapid cooling [10,15,16]. Generally, HLB value indicates the affinity of the surfactant for hydrophilic or lipophilic phases therefore an optimum HLB value and an optimum surfactant concentration play a crucial role in physical characteristics of nanoemulsions fourmulated by PIT method. Besides, the hydrophilicity of the emulsions system might be altered by differentiating compositions and co-surfactants [10,17]. Therefore, HLB value (*X*_1_), PEG-7 glyceryl cocoate (*X*_2_), and SOR (*X*_3_) were studied for optimizing their effects on three main characteristics of nanoemulsions including droplet size, PDI, and zeta potential.

The nanosized droplet is an attractive property of nanoemulsions leading to a pleasant transparent appearance and an excellent stability profile [13,14]. Likewise, PDI ranging from 0 to 1 is typically utilized for indicating uniformity and quality of the dispersion system, even nanoemulsions. A low PDI means low deviation of average size, which has shown a good uniformity of the system [9]. From the results, the increasing of PEG-7 glyceryl cocoate (HLB 11.0), which showed high affinity for the aqueous phase, leads to an increasing of the system hydrophilicity. Consequently, the emulsion system favored a higher HLB value of the surfactant. This finding also suggested that the smallest droplet size and PDI could be produced using certain conditions, consisting of the highest concentration of PEG-7 glyceryl cocoate (2%) and HLB value at the high level. Ostertag et al. also stated that the surfactant, presenting the lowest HLB value, produced the largest droplet size [14] which was in a good agreement with our results. According to the differences of HLB values, the phase behavior of the emulsion system at bicontinuous microemulsion phase and coalescence stabilization of the surfactant are altered which primarily influence the quality of nanoemulsions [14,29]. In addition, it is worth noting that surfactant concentration played a crucial role in the determination of droplet size and PDI. A number of studies also reported whether the decreasing of droplet size resulted from the increasing of surfactant concentration (high SOR) corresponding to our results [17,30]. This finding could be interpreted that at high concentration of surfactant, the bicontinuous microemulsion was allowed to be completely formed at the PIT point resulting in the subsequent lowest interfacial tension of the system, which facilitated droplet breakup [17]. Moreover, sufficient surfactant molecules were principally required to govern all small droplets after rapid cooling from the PIT [17,30].

Typically, zeta potential is one of the important parameters suggesting the stability of nanoemulsions [9]. Zeta potential represents an electrostatic repulsion of the similar charge between internal droplets resulting from the electrical double layer [31,32]. Generally, the decreasing of zeta potential indicates that the degree of repulsion was increased contributing to a good stabilization from coalescence [31]. In contrast to our results, the obtained nanoemulsions formulated using high levels of HLB and surfactant concentration, exhibited high zeta potential value with very small droplet size and narrow size distribution due to a low PDI, which tend to show a very good stability. This finding could be interpreted by the usage of non-ionic surfactants, which mainly relies on a steric stabilization between the surfactant molecules adsorbing onto the surface of droplets creating repulsive force rather than electrostatic force [33,34].

The optimum nanoemulsion formulation was subsequently incorporated with the extract. Noticeably, higher concentration of surfactant could produce a better homogenous system after rapid cooling compared to lower surfactant concentration [30]. After loading the extract EF into the plain nanoemulsions (N1 and N2), the droplet sizes of the formulations (N1-EF and N2-EF) were significantly increased indicating the successful encapsulation without the alteration of zeta potential. Furthermore, from our preliminary solubility study, the EF exhibited a lipophilic nature with water solubility of 0.0006 mg/L. Therefore, the EF preferably resided in the internal oil droplet rather than a watery continuous phase. However, in consideration of lower surfactant concentration of N1-EF, the unstable appearance then occurred after storage which might be explained by the principle of the PIT method. During storage at high temperature, the affinity of the surfactant for hydrophilic and lipophilic phases was altered resulting in the reduction of interfacial tension which promoted a subsequent merging of the droplets resulting in coalescence [17,35]. Consequently, larger droplet size, broader PDI, and visual unstability in the formulation containing lower surfactant concentration were observed. Nonetheless, the surfactant concentration of 9.5%w/w used in N2-EF could successfully stabilize the system during storage at high temperature. Notwithstanding, our results suggested that nanoemulsions could improve the chemical stability of the extract through encapsulation due to the protection of gallic acid from heat degradation. The outstanding anti-*P. acnes* and anti-inflammatory effects of the EF enables us to develop the nanoemulsions to enhance stability and skin permeation of the extract. The results also showed that nanoemulsions successfully enhanced the stability of antibacterial effect of the extract against *P. acnes* due to the encapsulation. Likewise, Ganesan et al., denoted that the stability of Achyrocline extract and lutein could be improved by encapsulation within nanoemulsions droplets [12].

One of the potential efficiencies of nanoemulsions is permeation enhancement through the primary skin barrier [36]. The recent hypothesis denoted that *P. acnes* typically resides in deep parts of the hair follicles [37,38]. Additionally, a variety of immunologic cells including monocytes, CD4+ T-cells, and Langerhans cells, which are significantly involved in the initiation of acne inflammation, are mainly found in the deeper epidermis dermis layer as well as in the vascular layer [39]. From these reasons, the active compound should be delivered through the stratum corneum and reach the VED and RC layer at which the target site of acne inflammation situates. Due to the fact that the very small droplet size of nanoemulsions (20–30 nm) provided enormous surface area, the extract was therefore allowed a relatively higher permeability through the skin [30]. The results showed that the nanoemulsions successfully facilitated the delivery of the extract through the SC. In addition, the hydrophilic part of nanoemulsions can hydrate the SC probably facilitating skin uptake of the compound [40]. Our findings suggested that the evolved nanoemulsions could successfully improve stability and skin permeability of *M. indica* kernel extract, which might provide better outcomes for topical anti-acne applications.

## 5. Conclusions

Our study firstly fabricated safflower-oil based nanoemulsions through using the PIT method together with optimized effects of HLB, PEG-7 glyceryl cocoate, and SOR on the main characteristics of nanoemulsions by CCD. Nanoemulsions, with very small droplet sizes and narrow size distribution, were certainly obtained using the optimum system conditions and compositions. Our results also demonstrated that the physicochemical and antibacterial stability as well as skin permeability of EF extract could be successfully improved by loading into the nanoemulsions fabricated using 9.5% of nonionic surfactant mixture. Therefore, *M. indica* kernel extract-loaded nanoemulsions could be a promising delivery system for anti-acne cosmeceutical products. However, skin irritation and performance testing in human volunteers are necessary for further investigation.

## Figures and Tables

**Figure 1 pharmaceutics-12-00454-f001:**
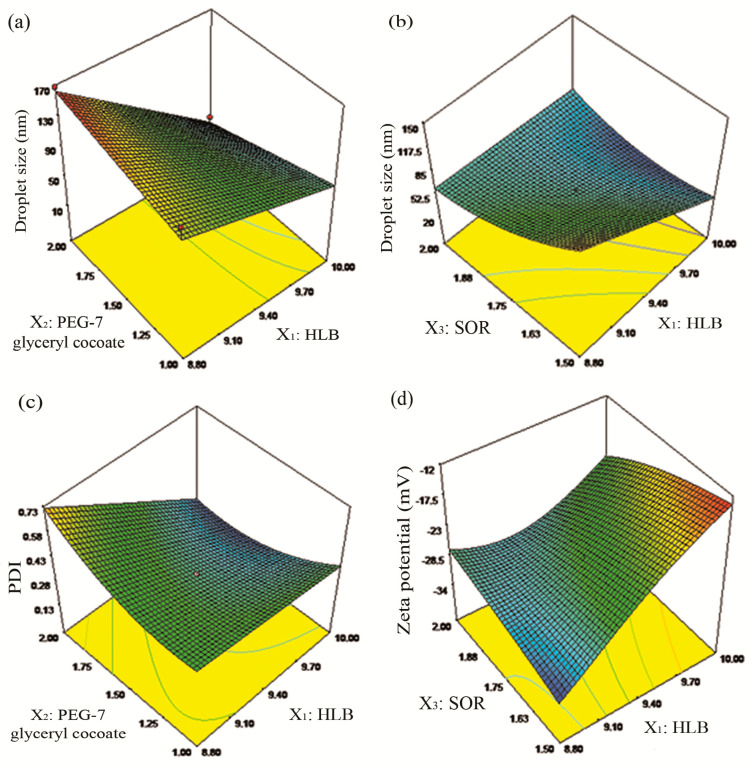
Response surface plots demonstrating the significant (*p* < 0.05) interaction effects on the studied variations including (**a**,**b**) droplet size (*Y*_1_: nm), (**c**) polydispersity index (PDI: *Y*_2_), and (**d**) zeta potential (*Y*_3_: mV).

**Figure 2 pharmaceutics-12-00454-f002:**
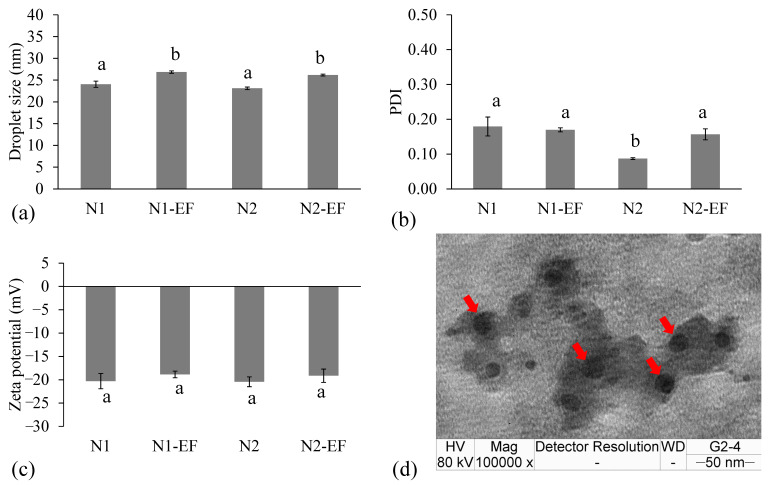
Mean droplet size (**a**), polydispersity index (PDI) (**b**) and zeta potential (**c**) of ethanolic fraction (EF)-encapsulated nanoemulsions (N1-EF and N2-EF) and their unloaded nanoemulsions (N1 and N2) produced by PIT method (mean ± SD; *n* = 3) together with ultrastructural image of EF-loaded nanoemulsions visualized through TEM (**d**). Different superscript letters (^a^ and ^b^) indicate the significant differences between groups using one-way ANOVA with multiple comparisons using Tukey (*p* < 0.05).

**Figure 3 pharmaceutics-12-00454-f003:**
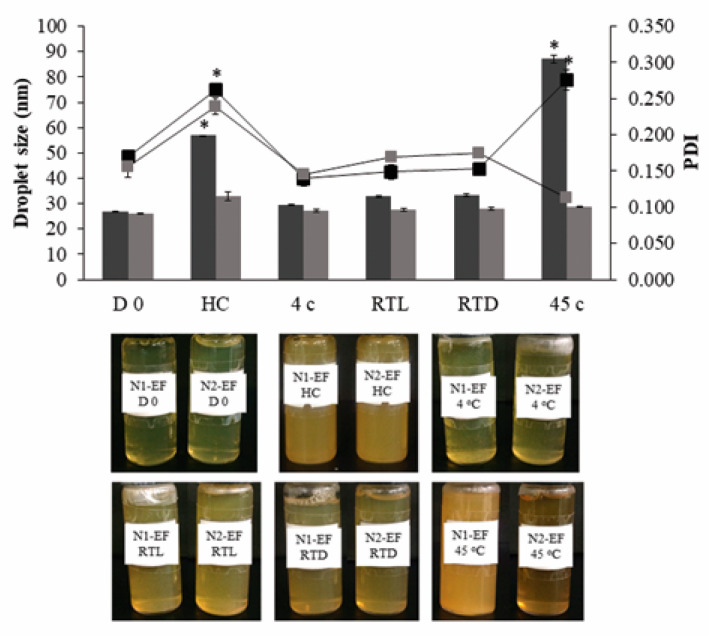
Mean droplet size and mean polydispersity index (PDI) of (■) N1-EF and (■) N2-EF at the starting point (D 0) and after storage in various conditions including heating-cooling (HC) for six cycles, 4 °C, room temperature with light (RTL), room temperature without light (RTD) and 45 °C for 90 d (mean ± SD; *n* = 3), * indicates a significant difference (*p* < 0.05) from D 0 of each formulation.

**Figure 4 pharmaceutics-12-00454-f004:**
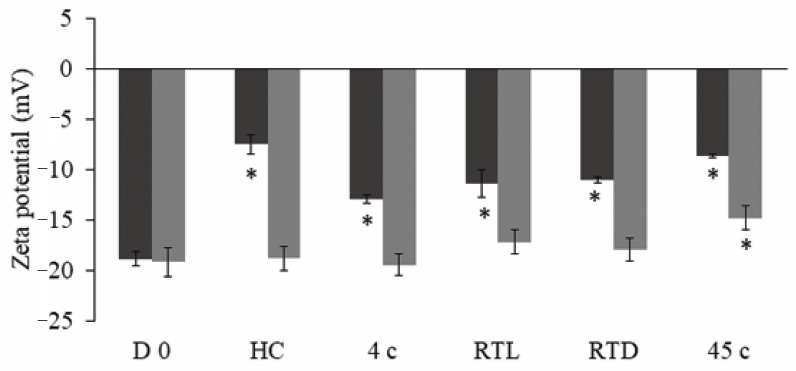
Mean zeta potential of N1- EF (■) and N2-EF (■) at the starting point (D 0) and after storage in various conditions including HC—heating-cooling for six cycles, 4 °C, room temperature with light (RTL), room temperature without light (RTD), and 45 °C for 90 d (mean ± SD; *n* = 3), * indicates a significant difference (*p* < 0.05) from D 0 of each formulation.

**Figure 5 pharmaceutics-12-00454-f005:**
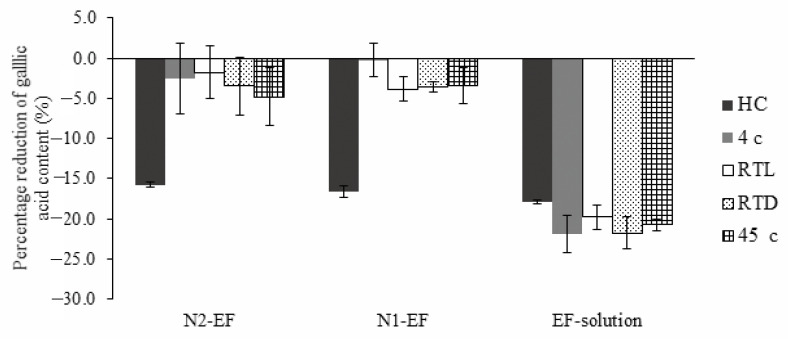
Percentage reduction of gallic acid content from the initial point (D0) of extract-loaded nanoemulsions (N1-EF and N2-EF) and the extract solution (EF-solution) evaluated using HPLC analysis after storage in various conditions (mean ± SD; *n* =3).

**Figure 6 pharmaceutics-12-00454-f006:**
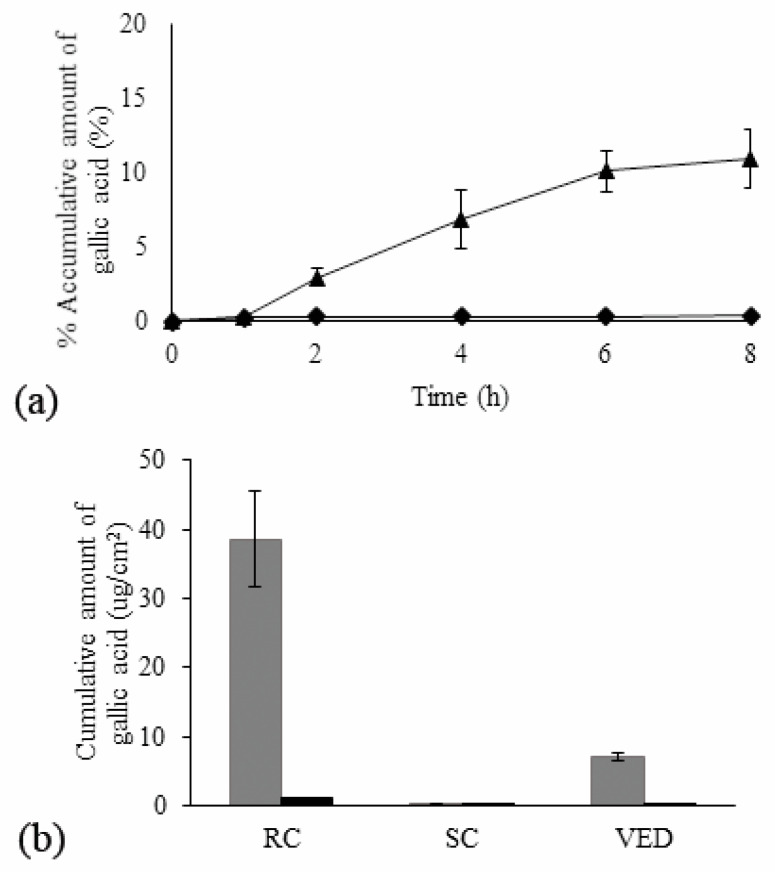
Percentage accumulative of gallic acid in receiving compartment (**a**); (
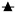
) N2-EF and (
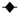
) EF-solution along with cumulative amount of gallic acid through skin per unit surface area after 8 h (*Q*_8_) in different layers (**b**); RC—receiver compartment, SC—stratum corneum, and VED—viable epidermis dermis; (■) N2-EF and (■) EF-solution (mean ± SD; *n* = 3).

**Table 1 pharmaceutics-12-00454-t001:** Levels of independent variables created according to the central composite design.

Independent Variables	Code Level				
	Axial (−α)	Low	Center	High	Axial (+α)
HLB value (*X*_1_)	8.40	8.8	9.4	10.0	10.4
PEG-7 glyceryl cocoate (*X*_2_: %)	0.66	1	1.5	2	2.34
Surfactant-to-oil ratio: SOR (*X*_3_)	1.33	1.5	1.75	2	2.17

**Table 2 pharmaceutics-12-00454-t002:** The matrix of central composite design (CCD) and the experimental data obtained for the response variables studied; *Y*_1_: droplet size, *Y*_2_: polydispersity index (PDI), and *Y*_3_: zeta potential (mean ± SD).

Run	Block	Independent Variables			Response Variables		
		*X* _1_	*X* _2_	*X* _3_	Droplet Size (*Y*_1_: nm)	Polydispersity Index (*Y*_2_)	Zeta Potential (*Y*_3_: mV)
1	1	9.4	0.66	1.75	57.3 ± 0.4	0.49 ± 0.01	−28.6 ± 0.8
2	1	10.0	2.0	2.00	17.8 ± 0.8	0.13 ± 0.01	−26.4 ± 2.9
3	1	9.4	1.5	1.75	52.3 ± 0.6	0.33 ± 0.20	−26.8 ± 0.5
4	1	9.4	1.5	2.17	69.2 ± 0.9	0.26 ± 0.04	−24.2 ± 1.3
5	1	10.4	1.5	1.75	21.2 ± 1.0	0.14 ± 0.01	−12.3 ± 0.4
6	1	10.0	2.0	1.50	22.9 ± 0.6	0.10 ± 0.02	−13.1 ± 2.9
7	1	9.4	1.5	1.75	52.3 ± 0.6	0.33 ± 0.20	−26.8 ± 0.5
8	1	10.0	1.0	2.00	94.8 ± 0.5	0.33 ± 0.16	−24.1 ± 0.7
9	1	8.8	1.0	2.00	55.6 ± 0.7	0.39 ± 0.09	−28.9 ± 0.1
10	1	9.4	1.5	1.75	52.3 ± 0.6	0.33 ± 0.20	−26.8 ± 0.5
11	1	8.4	1.5	1.75	97.9 ± 0.4	0.50 ± 0.01	−25.4 ± 0.3
12	1	8.8	2.0	2.00	86.2 ± 0.2	0.61 ± 0.01	−33.3 ± 0.2
13	1	9.4	1.5	1.75	52.3 ± 0.6	0.33 ± 0.20	−26.8 ± 0.5
14	1	9.4	1.5	1.75	52.3 ± 0.6	0.33 ± 0.20	−26.8 ± 0.5
15	1	9.4	1.5	1.33	121.3 ± 1.1	0.79 ± 0.09	−19.9 ± 0.5
16	1	9.4	1.5	1.75	52.3 ± 0.6	0.33 ± 0.20	−26.8 ± 0.5
17	1	9.4	2.34	1.75	71.4 ± 0.5	0.90 ± 0.10	−17.5 ± 0.7
18	1	8.8	2.0	1.50	167.3 ± 7.1	0.85 ± 0.27	−30.7 ± 0.5
19	1	10.0	1.0	1.50	217.0 ± 12.1	1.00 ± 0.01	−22.2 ± 0.5
20	1	8.8	1.0	1.50	137.6 ± 5.5	0.69 ± 0.04	−35.1 ± 0.2

**Table 3 pharmaceutics-12-00454-t003:** Non-ionic surfactant systems including Cetereth-20, PEG-40 castor oil, and sorbitan oleate varied by ratio for modifying five different hydrophile-lipophile balance (HLB) values.

Surfactant Systems	Ratio of Cetereth-20:PEG-40:Sorbitan Oleate	HLB Value
1	1:2:4	8.4
2	1:3.75:5	8.8
3	1:4:4	9.4
4	1:2:2	10.0
5	7:5:4	10.4

**Table 4 pharmaceutics-12-00454-t004:** Regression coefficients, *R*^2^, adjusted *R*^2^, and probability values for the final reduced equation.

Regression Coefficient	Droplet Size (*Y*_1_, nm)	PDI (*Y*_2_)	Zeta Potential (*Y*_3_, mV)
β _0_	59.27	0.34	−0.040
*X* _1_	−27.40	−0.16	−0.013
*X* _2_	−2.66	0.014	-
*X* _3_	−16.63	−0.075	−0.006
*X* _1_ ^2^	-	-	−0.006
*X* _2_ ^2^	-	0.094	-
*X* _3_ ^2^	15.72	-	-
*X* _12_	−22.57	−0.12	-
*X* _13_	23.34	-	0.009
*X* _23_	-	-	-
*R* ^2^	0.938	0.883	0.836
Adjusted *R*^2^	0.907	0.839	0.789
Regression (*p*-value)	<0.0001 ^a^	<0.0001 ^a^	<0.0001 ^a^

β_0_ is a constant; *X*_1_, *X*_2_, and *X*_3_ are the estimated regression coefficients for the main linear effects; *X*_1_^2^, *X*_2_^2^, and *X*_3_^2^ are the estimated regression coefficients for the quadratic effects; *X*_12_, *X*_13_, and *X*_23_ are the estimated regression coefficients for the interaction effects. *X*_1_: HLB, *X*_2_: PEG – 7 glyceryl cocoate (%), *X*_3_: Surfactant-to-oil ratio (SOR). ^a^ indicates a significant term (*p* < 0.05).

**Table 5 pharmaceutics-12-00454-t005:** The significance probability (*p*-value, *F*-value) of regression coefficients in the final reduced models.

Types of Effects	Variables	Droplet Size (*Y*_1_)		PDI (*Y*_2_)		Zeta Potential (*Y*_3_)	
		*F*-Value	*p*-Value	*F*-Value	*p*-Value	*F*-Value	*p*-Value
Main effects	*X* _1_	61.48	<0.0001	45.46	<0.0001	46.90	<0.0001
	*X* _2_	0.58	0.462 ^b^	0.35	0.5656 ^b^	-	-
	*X* _3_	22.64	0.0005	10.49	0.0065	10.60	0.0058
Quadratic effects	*X* _1_ ^2^	-	-	-	-	13.55	0.0025
	*X* _2_ ^2^	-	-	19.55	0.0007	-	-
	*X* _3_ ^2^	21.77	0.0007	-	-	-	-
Interaction effects	*X* _12_	22.33	0.0005	13.57	0.0028	-	-
	*X* _13_	23.89	0.0004	-	-	13.87	0.0023
	*X* _23_	-	-	-	-	-	-

^b^ indicated not significant at *p* > 0.05.

**Table 6 pharmaceutics-12-00454-t006:** The theoretical predicted and actual experimental response values of the optimum nanoemulsions system (*n* = 3).

Response	Predicted Value	Actual Value	%Prediction Error
Y_1_: Droplet size (nm)	20	20.78	3.75
Y_2_: Polydispersity index (PDI)	0.12	0.12	0
Y_3_: Zeta potential (mV)	−20.47	−21.14	3.17

**Table 7 pharmaceutics-12-00454-t007:** Minimum inhibitory concentration (MIC) and minimum bactericidal concentration (MBC) values (mg/mL) of the extract-loaded nanoemulsions (N1-EF and N2)-EFand the extract solution (EF-solution) after storage in various conditions for 90 d.

Storage Condition	N1-EF		N2-EF		EF-Solution	
	MIC	MBC	MIC	MBC	MIC	MBC
D 0	3.13	12.50	3.13	12.50	1.56	12.50
HC	3.13	12.50	3.13	12.50	3.13	12.50
4 °C	3.13	12.50	3.13	12.50	3.13	12.50
RTL	3.13	12.50	3.13	12.50	3.13	12.50
RTD	3.13	12.50	3.13	12.50	3.13	12.50
45 °C	3.13	12.50	3.13	12.50	3.13	12.50

Storage conditions: 4 °C, room temperature with light (RTL), room temperature without light (RTD), 45 °C and heating-cooling (HC).

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
