# Peer review of "Optimization of Mangifera indica L. Kernel Extract-Loaded Nanoemulsions via Response Surface Methodology, Characterization, Stability, and Skin Permeation for Anti-Acne Cosmeceutical Application"

_pharmaceutics, 2020, doi:10.3390/pharmaceutics12050454_

Round 1

Reviewer 1 Report

The reviewed manuscript entitled: “ Optimization of Mangifera indica L. Kernel Extract loaded Nanoemulsions via Response Surface Methodology, Characterization, Stability and Skin Permeation for Anti-acne Cosmeceutical Application” deals with optimization study for preparation of nanoemulsion formulations of plant antioxidative extract for targeted anti-acne cosmeceutical application. This study has deeply investigated the different effects as primary/secondary non-ionic surfactants/oils ratios, the use of rapid cooling the primary emulsion and its effect of the final nano emulsion formation etc. with a final result – well-stable nanoemulsion plant extract formulations which sounds promising for their future biocosmetic and why not for a future biomedical/drug delivery application.

For Optimization of the nanoemulsions formulation, the experimental design was carried out based to PIT method (Phase inversion temperature – should be mentioned first, before abbreviated!). As the PIT method relies on changing of the temperature which alters an affinity of thermal-sensitive non-ionic surfactants particularly polyethoxylated surfactants, for hydrophilic and lipophilic phases, I would recommend the authors in their future studies to try to experiment with some more temperature-sensitive polymers like the commercial Pluronic family (polypropylene/ethylene oxide co-polymers) or the Tyloxapol non-ionic surfactant.

2.3. Preparation of nanoemulsions. This part could be improved with more detailed description as it is not clear what happens with the ethanol from the ethanolic plant extract during emulsions formulation – does it evaporate or remain and at what quantity? What is the extracted fraction in wt.%, the dry content? Also, what BHT stands for?

Author Response

When we have got your valuable suggestions and comments which provided us an opportunity to improve and correct this manuscript, we carefully re-checked and corrected all issues following your instructions. Those corrected issues were clarified point-by-point and explained as described as follows:

Point 1: For Optimization of the nanoemulsions formulation, the experimental design was carried out based to PIT method (Phase inversion temperature – should be mentioned first, before abbreviated!). As the PIT method relies on changing of the temperature which alters an affinity of thermal-sensitive non-ionic surfactants particularly polyethoxylated surfactants, for hydrophilic and lipophilic phases, I would recommend the authors in their future studies to try to experiment with some more temperature-sensitive polymers like the commercial Pluronic family (polypropylene/ethylene oxide co-polymers) or the Tyloxapol non-ionic surfactant.

Responses 1 : Phase inversion temperature as a full name of “PIT” was already mentioned at Line 59, Page 2 in the introduction part which is the first location of this term.

Also, we’re very grateful for your valuable suggestion regarding promising temperature-sensitive polymers for our further experiment. We will conduct these materials for sure.

Point 2 : 2.3. Preparation of nanoemulsions. This part could be improved with more detailed description as it is not clear what happens with the ethanol from the ethanolic plant extract during emulsions formulation – does it evaporate or remain and at what quantity?

Responses 2 : Corrected – “Ethanol was totally excluded through evaporation ..” was added in Section 2.2., Line 93, Page 3 to clarify that the ethanol used as an extracting solvent was evaporated until the formation of dark brown semi-solid extract without any remaining ethanol and did not interfere the nanoemulsions system, as well. Extract’s appearance was also denoted in Section 2.2., Line 94-95, Page 3.

Point 3 : What is the extracted fraction in wt.%, the dry content?

Responses 3 : Corrected – “with a yield of 12.12 ± 1.19 % w/w dry weight” has been already added into Section 2.2., Line 95, Page 3.

Point 4 : Also, what BHT stands for?

Responses 4 : Corrected - Butylated hydroxytoluene as a full name of “BHT” which served as a lipophilic antioxidant in nanoemulsions for preventing possible oxidation in the system has been already added into Section 2.3., Line 100, Page 3.

            We truly appreciated for your suggestions and recommendations. All these explanations were enclosed in an attachment file.

Sincerely Yours,

Authors

Reviewer 2 Report

The present manuscript deals with the use of nanoemulsions to increase stability and skin permeation of Mangifera Indica kernel extract. The authors characterised the loaded nanoemulsion after a preliminary optimization of the formulation by response surface methodology. The experimental work is quite well organised and the manuscript readable. Some points should be addressed.

Line 65-67 Please rephrase

Paragraph 2.2. It is not clear the extraction procedure. At the end of maceration ethanol was evaporated. How ethyl acetate was removed? At the end of the procedure, what it is obtained, a solution or a liquid/solid mixture?

Line 99 How was determined the PIT temperature of surfactant mixtures?

Please define α in the central composite design

Paragraph 2.6 It is not clear how the extract was loaded in the nanoemulsion. Is the extract lipophilic or hydrophilic? Is it really encapsulated in the droplets of the inner phase or is solubilised in the external aqueous phase? This is a crucial point. Please reorganised this paragraph, by explaining how the extract was loaded effectively in the nanoemulsion.

Line 211 What is “for window”?

Paragraph 3.2 Results from the optimization of the responses should be shown.

Line 265-266 This sentence is not clear

Line 307 table 6 is not in the manuscript

Line 355 and 395 References are not well formatted

Author Response

When we have got your valuable suggestions and comments which provided us an opportunity to improve and correct this manuscript, we carefully re-checked and corrected all issues following your instructions. Those corrected issues were clarified point-by-point and explained as described as follows:

Point 1: Line 65-67 Please rephrase

Responses 1 : Corrected – “The development of these statistic models potentially indicated the significance of the independent variables and the investigation of synergistic or antagonistic interactions between different studied variables”, this statement has been included instead of the former phrase as shown in Line 65-67, Page 2.

Point 2 : Paragraph 2.2. It is not clear the extraction procedure. At the end of maceration ethanol was evaporated. How ethyl acetate was removed?

Responses 2 : Regarding extraction, two solvents including ethyl acetate and 95% ethanol were employed after de-waxing for fractionation. The dewaxed plant materials were firstly macerated with ethyl acetate as a procedure described in the manuscript. The ethyl acetate filtrate was collected and evaporated for excluding the solvent. Subsequently, plant residue was fractionated by 95% ethanol for 48 h, 3 cycles and ethanol filtrate was collected and totally excluded through evaporation to give the ethanolic fraction which was utilized as an active ingredient herein.

            Therefore, in order to clarify this issue, the passage in Section 2.2., Line 91-95, Page 3, has been rearranged into “The plant residue was then fractionally macerated using ethyl acetate for 48 h, 3 cycles. The plant residue from ethyl acetate fractionation was then macerated followed by 95% ethanol for 48 h, 3 cycles. Ethanol was evaporated totally excluded through evaporation using rotary evaporator (Buchi® Rotavapor R-300, Thailand) to produce an ethanolic fraction (EF)with a yield of 12.12 ± 1.19 %w/w dry weight”.

Point 3 : At the end of the procedure, what it is obtained, a solution or a liquid/solid mixture?

Responses 3 : The appearance of final extract was dark-brown, semi-solid. We also stated this explanation in Section 2.2., Line 94-95, Page3.

Point 4 : Line 99 How was determined the PIT temperature of surfactant mixtures?

Responses 4 : In our preliminary study, the PIT temperatures of surfactant mixtures were determined following the method of Rao et al., (2010) as conductivity measurement. The mixture of surfactant (10%), oil phase (5%) and DI water (85%) was heated at a controlled rate and concurrently measured the alteration of electrical conductivity. At PIT temperature, the conductivity sharply decreased implying phase inversion from O/W emulsion (representing high conductivity) into W/O emulsion of which oil is a continuous phase. Our preliminary results thus provided the PIT temperature used in this study.

Therefore, a phrase “that was previously determined by conductivity measurement, as 78.5 oC” has been added in Section 2.3, Line 101-102, Page 3 to clarify this issue.

Reference : Rao J, McClements DJ. Stabilization of phase inversion temperature nanoemulsions by surfactant displacement. J Agric Food Chem 2010; 58: 7059-66.

Point 5 : Please define α in the central composite design

Responses 5 : Regarding to central composite design, “α” technically represent axial point of the design allowing estimation of curvation. Also, factorial design with center point generally embedded together with the axial points. Sometimes, a term of “star point” was used instead but α or axial point is the most common term. As shown in Table 1, axial point has been already stated and α is a term of its abbreviation.

Point 6 : Paragraph 2.6 It is not clear how the extract was loaded in the nanoemulsion. Is the extract lipophilic or hydrophilic? Is it really encapsulated in the droplets of the inner phase or is solubilised in the external aqueous phase? This is a crucial point. Please reorganised this paragraph, by explaining how the extract was loaded effectively in the nanoemulsion.

Responses 6 : From our preliminary determination, solubility of the extract was determined. Our result exhibited that this extract was a lipophilic compound with a water solubility of 0.0006 mg/l. Therefore, this lipophilic substance preferably resides in internal oil phase indicating the encapsulation within the nanoemulsions internal droplet.

Consequently, in order to clarify this issue and confirm the effective encapsulation, “Furthermore, from our preliminary solubility study, the EF exhibited a lipophilic nature with water solubility of 0.0006 mg/l. Therefore, the EF preferably resided in the internal oil droplet rather than a watery continuous phase” has been added into Discussion part, Section 4, Line 398-400, Page 13.

Point 7 : Line 211 What is “for window”?

Responses 7 : A term “for window” herein defines the SPSS software version 17.0 designed for Window Operating System. However, because “window” is a name, “W” uppercase was used instead as shown in Line 217, Page 6.

Point 8 : Paragraph 3.2 Results from the optimization of the responses should be shown.

Responses 8 : Corrected – A new Table was added into the manuscript demonstrating the optimization of the responses and assigned to be Table 6 as shown in Section 3.2., Line 274, Page 8. Also, the former Table 6 was changed into Table 7 in the revised manuscript.

Point 9 : Line 265-266 This sentence is not clear

Responses 9 : Percentage prediction error is an element for verification of the model. Percentage prediction errors less than 5% suggested that percent change of theoretical predicted values from actual experiment value is acceptable.

Corrected – “The final models were acceptable” has been added into Section 3.2., Line 272, Page 8.

Point 10 : Line 307 table 6 is not in the manuscript

Responses 10 : We truly apologized for this shortcoming. Since, the new Table 6 has been added into Section 3.2., Page 8. Consequently, the former Table 6 has been assigned to be Table 7 and already added into Section 3.4., Page 11.

Point 11 : Line 355 and 395 References are not well formatted

Responses 11 : Corrected –  These two references have been corrected as shown in Line 371 -372 and Line 413-414, respectively. (Page 13)

            We truly appreciated for your suggestions and recommendations. These explanation were enclosed in an attachment file.

Sincerely Yours,

Authors

Reviewer 3 Report

Authors Worrapan Poomanee  et al made an attempt to prepare nano-formulation of mangifera indica kernel extract and characterize it's stability and skin permeability. 

Authors have done extensive experiments for their study and the manuscript is written appropriately. Authors have done formulation, characterization, Stability of the formulation at different conditions followed by in vitro and in situ studies for efficacy. 

Please address the following comments

  1. Line # 93 - Please mention the physical appearance of the extract.
  2. Line # 126, 128 - Table 1 & Table 2. Please check the parenthesis symbols or explain
  3. Line # 146 - please provide reference or rewrite the sentence
  4. Line # 154 - Can you provide TEM images which will improve the quality of the manuscript
  5. Line # 185: For stability study was there any positive control used for comparison. please provide information
  6. Line # 205 - Eq 5. please mention as A%
  7. Line # 307 indicates table 6 results, but main manuscript do not have table 6.
  8. Appendix has duplicate of all the results (Except table 6)
  9. Line # 275 - Fig. 2b y axis scale may be decreased to have better bars
  10. conclusion: please mention which percentage of surfactant is good and which one you would suggest for next level study i.e. clinical study
  11. Please reduce the discussion and focus towards your results mainly (if possible).

Author Response

When we have got your valuable suggestions and comments which provided us an opportunity to improve and correct this manuscript, we carefully re-checked and corrected all issues following your instructions. Those corrected issues were clarified point-by-point and explained as described as follows:

Point 1: Line # 93 - Please mention the physical appearance of the extract.

Responses 1 : Corrected – “as a dark brown semi-solid extract” has been already added into Section 2.2., Line 94-95, Page 3.

Point 2 : Line # 126, 128 - Table 1 & Table 2. Please check the parenthesis symbols or explain

Responses 2 : Corrected – We have already checked thoroughly the symbols. Also, Abbreviation of each response variables were clarified in Title of Table 2 as shown in the revised manuscript.

Point 3 : Line # 146 - please provide reference or rewrite the sentence

Responses 3 : Corrected – The sentence has been changed into “The N1-EF and N2-EF were prepared by the PIT method as previously described in Section 2.3.”  as shown in Section 2.6., Line 152, Page 4.

Point 4 : Line # 154 - Can you provide TEM images which will improve the quality of the manuscript

Responses 4 : The first version of this manuscript has already provided a TEM image as shown in Figure 2d. We added a new version of Figure 2 in Line 285, Page 9 with additional red arrows to intensify the presence of droplets. The indication of TEM image was denoted in Section 3.3., Line 281-282, Page 9 as well as the title of Figure 2, Line 290, Page 9.

Point 5 : Line # 185: For stability study was there any positive control used for comparison. please provide information

Responses 5 : Regarding our stability experiment, the extract-loaded nanoemulsions (N1-EF and N2-EF) were compared to the extract (EF)-solution (extract dissolved in PEG-400) to study the ability of nanoemulsions in stability enhancement which was compared to a solution form.

For chemical stability determination, gallic acid served as a reference standard for clarifying the degradation of the extract. Also, for antibacterial stability, gallic acid dissolved in 15% ethanol in broth also served as a positive control in the MIC/MBC experiments. Our previous experiment showed that 15% ethanol has no effect on P. acnes.

Point 6 : Line # 205 - Eq 5. please mention as A%

Responses 6 : Corrected – A% has been added instead as shown in Section 2.10., Line 210, Page 6.

Point 7 : Line # 307 indicates table 6 results, but main manuscript do not have table 6.

Responses 7 : We truly apologized for this shortcoming. Nevertheless, due to the addition of new Table 6 in Section 3.2., Page 8, Table 7 describing the information shown in former Line 307 has been already added into Section 3.4., Page 11 for describing the information shown in Line 317-322, Page 10-11.

Point 8 : Appendix has duplicate of all the results (Except table 6)

Responses 8 : Corrected – A new Table 6 and Table 7 have been already added. Also, the presence of Appendix has followed the Journal format file.

Point 9 : Line # 275 - Fig. 2b y axis scale may be decreased to have better bars

Responses 9 : In case of graph illustrating Polydispersity index (PDI), the range of 0.0 – 0.5 could clearly clarify the small value of this term which indicates a desirable nanoemulsions with narrow size distribution.

Point 10 : conclusion: please mention which percentage of surfactant is good and which one you would suggest for next level study i.e. clinical study

Responses 10 : Corrected – “fabricated using 9.5% of nonionic surfactant mixture” has been added into Section 5, Line 436, Page 14.

Point 11 : Please reduce the discussion and focus towards your results mainly (if possible).

Responses 11 : Supporting information described in this section were performed for comparison and discussion with our findings. We try to reduce some phrases as shown in Section 4.

We truly appreciated for your suggestions and recommendations. These explanation was enclosed in an attachment file.

Sincerely Yours,

Authors

Round 2

Reviewer 2 Report

The authors have addressed the comments. The manuscript is suitable for pubblicatiom.

Line 216 the operating system is "Windows" and not "Window"

Line 400 "storage" typo mistake